# Antiplanktonic and Antibiofilm Activity of *Rheum palmatum* Against *Streptococcus oralis* and *Porphyromonas gingivalis*

**DOI:** 10.3390/microorganisms10050965

**Published:** 2022-05-03

**Authors:** Nadine Kommerein, Nina Vierengel, Jonathan Groß, Till Opatz, Bilal Al-Nawas, Lena Katharina Müller-Heupt

**Affiliations:** 1Department of Prosthetic Dentistry and Biomedical Materials Science, Hannover Medical School, Carl-Neuberg-Straße 1, 30625 Hannover, Germany; kommerein.nadine@mh-hannover.de; 2Department of Chemistry, Johannes Gutenberg-University, Duesbergweg 10–14, 55128 Mainz, Germany; vierengel@uni-mainz.de (N.V.); jgross03@uni-mainz.de (J.G.); opatz@uni-mainz.de (T.O.); 3Department of Oral- and Maxillofacial Surgery, University Medical Center Mainz, Augustusplatz 2, 55131 Mainz, Germany; bilal.al-nawas@unimedizin-mainz.de

**Keywords:** antibiofilm, biofilm, antimicrobial, pathogen-selective therapeutics, periodontitis, peri-implantitis, planktonic, *Porphyromonas gingivalis*, *Rheum palmatum*, *Streptococcus oralis*

## Abstract

Periodontitis and peri-implantitis are inflammatory conditions with a high global prevalence. Oral pathogens such as *Porphyromonas gingivalis* play a crucial role in the development of dysbiotic biofilms associated with both diseases. The aim of our study was to identify plant-derived substances which mainly inhibit the growth of “disease promoting bacteria”, by comparing the effect of *Rheum palmatum* root extract against *P. gingivalis* and the commensal species *Streptococcus oralis*. Antiplanktonic activity was determined by measuring optical density and metabolic activity. Antibiofilm activity was quantified using metabolic activity assays and live/dead fluorescence staining combined with confocal laser scanning microscopy. At concentrations of 3.9 mg/L, *R. palmatum* root extract selectively inhibited planktonic growth of the oral pathogen *P. gingivalis*, while not inhibiting growth of *S. oralis*. Selective effects also occurred in mature biofilms, as *P. gingivalis* was significantly more stressed and inhibited than *S. oralis*. Our studies show that low concentrations of *R. palmatum* root extract specifically inhibit *P. gingivalis* growth, and offer a promising approach for the development of a potential topical agent to prevent alterations in the microbiome due to overgrowth of pathogenic *P. gingivalis*.

## 1. Introduction

Periodontitis and peri-implantitis remain global health problems with high prevalence rates across many countries [1,2]. According to analyses of the 2019 Global Burden of Disease Study, the global burden of severe periodontitis has substantially increased in recent decades [3]. Periodontitis is a disease of the gums surrounding the teeth, and is influenced by many interacting factors, including the host immune response, the inflammatory response and the subgingival microbes [4]. Patients with periodontitis are at increased risk of swollen or bleeding gums, bad breath, tooth loss and edentulism, which adversely impact oral related quality of life [5]. Furthermore, periodontitis is known to have a mutual impact on systemic health and may negatively influence diseases such as diabetes, the overall cancer risk or cardiovascular disease [6,7] and is a possible risk factor for the development of peri-implantitis. Peri-implantitis is defined as a biofilm-associated “pathological condition occurring in tissues around dental implants, characterized by inflammation in the peri-implant mucosa and subsequent progressive loss of supporting bone” [2]. The prevalence of peri-implantitis is relatively high, at 26% [8]. Oral pathogens and their activity in dysbiotic biofilms play a crucial role in the development of both diseases. A shift from subgingival Gram-positive bacteria to mainly Gram-negative anaerobic bacteria results in a dysbiotic oral biofilm [9,10]. Periodontitis and peri-implantitis pathogens involve different bacteria. Bacteria highly associated with periodontitis include *Porphyromonas gingivalis*, *Tannerella forsythia*, *Fusobacterium nucleatum*, *Treponema denticola* and *Aggregatibacter actinomycetemcomitans* [11,12,13]. In general, periodontal pathogens are found in lower concentrations in peri-implantitis, but *P. gingivalis*, *T. forsythia* and *F. nucleatum* are also abundant in peri-implantitis [14].

Current therapeutic strategies for periodontitis and peri-implantitis involve nonsurgical procedures, such as mechanical removal of supra- and subgingival biofilms, surgical procedures and the use of adjuvant antiseptics or antibiotics to reduce overall bacterial load [15,16]. These treatments change ecological conditions in the subgingival pockets [17], reduce inflammation and thus tend to suppress the ecological niche for the growth and re-growth of pathogenic bacteria [18]. If therapeutic strategies for these two diseases are to be effective, it is essential that they ultimately lead to the recovery of microbial homeostasis. Thus, antiseptics such as chlorhexidine (CHX) or povidone iodine (PVP-Iod) are commonly used as adjuvants in dentistry. Both CHX and PVP have broad spectra of antimicrobial activity against Gram-positive and Gram-negative bacteria [19,20]. Thus, CHX keeps the overall bacterial count low by unselective targeting of bacteria and therefore decreases the bacterial biomass. Nevertheless, unselective targeting of oral bacteria may facilitate re-infection by pathogens, since the resident microflora is then less diverse and there is then less competition for pathogens [21]. In short-term use, clinical studies have demonstrated a reduction in microbial diversity and a shift in the microbiome, leading to more acidic conditions and lower nitrite availability [22,23]. Therefore, the use of CHX may lead to less bacterial diversity. However, a systematic review could not be completed since more clinical studies are needed to conclude long-term positive or negative effects on oral microbiota [24]. Well known common side effects of oral CHX use are burning sensations in the mouth, mucosal irritation, increased tooth staining, increased calculus formation and altered taste perception [25,26].

For the above-mentioned reasons, novel therapeutic strategies such as the use of pre- or probiotics have been evaluated in order to rectify bacterial imbalance in periodontal or peri-implantitis therapy. Pre-clinical and in vitro studies have shown promising results for a positive shift in oral microbiota, but long-term effects are not yet fully elucidated [27,28]. To complement novel concepts addressing bacterial imbalance, there is a need for novel treatment options, which specifically target pathogens in oral biofilms without targeting commensal bacteria. *Rheum palmatum* root extract showed promising antibacterial results against planktonic *P. gingivalis*. A minimum inhibitory concentration (MIC) of 4 mg/L was reported using broth microdilution assay [29]. Kampo is a traditional Japanese herbal formulation containing *R. palmatum* rhizomes and *Glycyrrhiza* roots; a study by Liao et al. has shown that this inhibits on *P. gingivalis* growth and virulence properties, such as bacterial adherence to epithelial cells [30].

In order to identify plant-derived substances for long-term topical use in oral hygiene, which inhibit mainly the growth of “disease promoting bacteria”, we have now compared the effects of *R. palmatum* root extract on different oral bacteria, such as *P. gingivalis* and *Streptococcus oralis* [11]. *P. gingivalis* is a major periodontal pathogen and is associated with periodontitis and peri-implantitis. A healthy composition of the oral commensal microbiota can be changed and disturbed by *P. gingivalis*, even at very low colonisation levels [31], leading to disruption of the host-microbial homeostasis [32]. Streptococci make up over 80% of early commensal oral biofilms [33,34,35]. *S. oralis* is an oral commensal that is highly abundant in healthy dental implant sites [36] and has been shown to be capable of actively protecting oral tissue homeostasis in vitro [37]. 

Therefore, our study aimed to examine the effect of *R. palmatum* root extract on oral mature biofilms of these two bacterial species, since it is well known that bacteria in biofilms exhibit greater resistance to antiseptics than planktonic bacteria do [38].

## 2. Materials and Methods

### 2.1. Microorganisms and Culture Conditions 

*S. oralis* strain ATCC 9811 was obtained from the American Type Culture Collection (ATCC) and *P. gingivalis* strain ATCC 33277/DSM 20709 from the German Collection of Microorganisms and Cell Cultures (DSMZ). The strains were grown under anaerobic conditions (80% N_2_, 10% H_2_, 10% CO_2_; anaerobic workbench Concept 400-M, Ruskinn Technology Ltd., Pencoed, UK) at 37 °C for 24 h in 50 mL screw cap tube (Sarstedt, Nümbrecht, Germany) with Brain Heart Infusion Medium (BHI; Oxoid, Wesel, Germany) containing 10 µg/mL vitamin K (Roth, Karlsruhe, Germany; BHI/VitK) to obtain 24-h-old precultures for further processing. 

### 2.2. Preparation of Rheum Palmatum Extract Solutions

*R. palmatum* root extract was obtained from Paninkret (Paninkret, Pinneberg, Germany). The autoclaved substances were dissolved in sterile filtered, deionised and autoclaved water. By serial dilutions with water (for planktonic experiments) or BHI/VitK (for biofilm experiments), the solutions were adjusted to concentration of 1.95–1000 mg/L. Solutions were freshly prepared for each experiment. The identity of the *R. palmatum* root extract used in this study was identified according to the European Pharmacopeia and the European Union herbal monograph on *Rheum palmatum* L. and *Rheum officinale* Baillon, radix (EMEA/HMPC/189624/2007).

### 2.3. HPLC (High-Performance Liquid Chromatography) Analysis 

HPLC of *R. palmatum* root extract (Paninkret, Pinneberg, Germany) was performed on an Agilent Infinity II 1260 system with a diode array detector and a quadrupol ESI mass spectrometer. The results of the HPLC analysis containing the UV chromatogram at 254 nm and the corresponding peak list have been published in a previous study [29]. The UV chromatogram at 254 nm and the extracted ion current chromatograms (EIC) of the five most abundant anthraquinones and their glycosides in negative ionization mode (ESI–) aredepicted in Appendix A.

### 2.4. Investigation of the Planktonic Bacterial Growth and Metabolic Activity of S. oralis and P. gingivalis under the Influence of R. palmatum Root Extract

In order to determine the influence of *R. palmatum* root extract on planktonic *S. oralis* and *P. gingivalis*, the 24-h precultures were pelleted by centrifugation at 4000× *g* for 15 min at 4 °C, the supernatant was discarded and the bacterial pellets were resuspended in fresh doubly concentrated BHI/VitK medium (2× BH/VitK). The optical density was measured at 600 nm (OD_600_; BioPhotometer, Eppendorf, Hamburg, Germany), adjusted to 0.2 and diluted 1:10 with 2× BHI/VitK to 0.02. The freshly prepared *R. palmatum* root extract solutions were mixed equally with the bacterial cultures (OD_600_ = 0.02) to a final OD_600_ = 0.01, with the following final concentrations as depicted in Table 1.

As positive controls, bacterial cultures (OD_600_ = 0.02) were mixed 1:2 with sterile filtered, deionised and autoclaved water (growth controls). As negative controls, 2× BHI/VitK was mixed equally with sterile filtered, deionised and autoclaved water (medium control) and *R. palmatum* root extract solutions were mixed 1:2 with 2× BHI/VitK (*R. palmatum* solution control). Finally, 150 µL of each suspension was transferred into each well of a 96-well plate (Nucleon 96 Flat Bottom Transparent Polystyrene; Thermo Fisher Scientific, Waltham, MA, USA) and cultured for 24 h at 37 °C, with rotation (180 rpm; Shaking Incubator Typ 3032, GFL, Burgwedel, DE) under anaerobic conditions (Anaerobic Jar and AnaeroGen; Oxoid, Wesel, Germany). All experiments were carried out in three biological and two technical replicates.

To evaluate the effect of *R. palmatum* root extract on the growth of planktonic *S. oralis* and *P. gingivalis* cultures, the 24-h bacterial cultures were mixed and the optical density (OD_600_) was measured with a plate reader (Tecan, Mennedorf, Switzerland). Metabolic activity was measured using the BacTiter-Glo^TM^ Microbial Viability Assay (Promega, Mannheim, Germany). 50 µL BacTiter-Glo^TM^ reagent and 50 µL of the well mixed bacterial cultures were added to opaque 96-well plates (Nucleon 96 Flat Bottom Black Polystyrene; Thermo Fisher Scientific, Waltham, MA, USA). After 5 min incubation at room temperature under light protection, the samples were mixed again, and the amount of adenosine triphosphate (ATP) was determined by measuring the luminescence using the plate reader. All results were normalised to the medium or to the control *R. palmatum* root extract solution. 

### 2.5. Investigation of R. palmatum Root Extract Effects on S. oralis and P. gingivalis Biofilms 

To evaluate the effect of *R. palmatum* root extract on biofilms of *S. oralis* and *P. gingivalis*, 24-h-old precultures were processed as described for the planktonic experiments, except that 1× BHI/VitK medium was used and that OD_600_ was adjusted to 0.1. 2 mL of each bacterial culture (OD_600_ = 0.1) were added to each well of a 6-well plate (Cellstar; Greiner Bio-One, Frickenhausen, Germany) and cultured for 24 h at 37 °C under anaerobic conditions. All experiments were carried out in three biological and two technical replicates.

After 24 h of growth, supernatants were removed, 2 mL fresh BHI/VitK was added to the controls, and 2 mL of *R. palmatum* root extract solutions were added to the biofilms that were be treated with final concentrations of 1.95 mg/L, 31.25 mg/L, 500 mg/L.

After two hours of incubation under anaerobic conditions at 37 °C, the metabolic activity was determined using the BacTiter-Glo Microbial Viability Assay (Promega, Mannheim, Germany). Biofilms were rinsed twice with Phosphate Buffered Saline (PBS; Biochrom GmbH, Berlin, Germany) and 1 mL BacTiter-Glo reagent was added to each biofilm. The biofilms were rinsed with the reagent by pipetting up and down several times. After 5 min of incubation under rotation (180 rpm) and light protection, 100 µL of the solutions were added to opaque 96-well plates. Samples were mixed again and the luminescence was measured with the plate reader. In order to analyse the effect *R. palmatum* root extract on biofilm volume and live/dead distribution, the LIVE/DEAD BacLight Bacterial Viability Kit (Life Technologies, Carlsbad, CA, USA) was used to stain the biofilms with the green fluorescent nucleic acid stain SYTO 9 and the red fluorescent nucleic acid stain propidium iodide according to the manufacturer’s recommendations. Subsequently, the biofilms were rinsed twice with PBS and fixed with 2.5% glutardialdehyde (Carl Roth, Karlsruhe, Germany) in PBS for 30 min. After fixation, the biofilms were washed twice and covered with 3 mL PBS. They were then microscopically analysed using a confocal laser scanning microscope (CLSM; Leica TCS SP8, Leica Microsystems, Mannheim, Germany) with excitation and emission wavelengths for SYTO 9 of 488 nm and 500–550 nm and for propidium iodide of 552 nm and 675–750 nm. Three images were taken from each biofilm, using a z-step of 1 μm. Three-dimensional image processing and analysis of the biofilm volumes and proportions of viable (green; SYTO 9), dead (red; propidium iodide) and colocalized (orange; SYTO 9 + propidium iodide) bacteria was performed using Imaris x64 software (version 8.4.1, Bitplane AG, Zurich, Switzerland). Colocalized cells were defined as dead because the propidium iodide was able to penetrate the membrane.

### 2.6. Statistical Analysis

Graphic processing and statistical analysis were performed using the GraphPad Prism software 8.4 (GraphPad Software Inc., La Jolla, CA, USA). To determine whether the data are normally distributed, the Kolmogorov–Smirnov normality test was applied. If data were normally distributed, Ordinary One-Way ANOVA with Dunnett’s correction for multiple comparisons was used for to determine statistically significant differences of treated samples compared to the controls. If data failed the normal distribution assumption, the Kruskal–Wallis test with Dunn’s correction for multiple comparisons was used. The significance level was set to *p* ≤ 0.05 for all comparisons.

## 3. Results

### 3.1. Selective Antimicrobial Effects of R. palmatum Root Extract on Planktonic S. oralis and P. gingivalis

*R. palmatum* root extract significantly inhibited planktonic growth of *S. oralis*, starting at a concentration of 62.5 mg/L (Figure 1A). The metabolic activity of *S. oralis* increased significantly from 31.25 mg/L *R. palmatum* extract (except for 125 mg/L) (Figure 1C). Planktonic growth of *P. gingivalis* was already severely reduced at a concentration of 3.9 mg/L, and significant reduction started at 15.625 mg/L *R. palmatum* root extract (Figure 1C). The metabolic activity of *P. gingivalis* was likewise significantly reduced-starting at a concentration of 31.25 mg/L (Figure 1D).

### 3.2. Selective Antibiofilm Activity of R. palmatum Root Extract on S. oralis and P. gingivalis 

One representative CLSM image for each experiment of *S. oralis* and *P. gingivalis* biofilms after 24 h of cultivation followed by 2 h treatment with *R. palmatum* root extract is shown in Figure 2. With increasing concentration of *R. palmatum* root extract, the proportion of dead bacteria within the biofilms increased visually for both species; however, the antibacterial effect seemed to be much stronger for *P. gingivalis* than for *S. oralis*. For both bacterial species, the vertical sections showed a considerable increase in biofilm thickness and the biofilm structures became more porous and holey, as the concentration of *R. palmatum* increased, in particular at 500 mg/L. 

The biofilm volumes of *S. oralis* increased slightly under the influence of *R. palmatum* root extract compared to the control; at a concentration of 31.25 mg/L, the increase was significant (Figure 3A). The proportion of viable bacteria was significantly lower under the influence of *R. palmatum* extract than for the control biofilms (Figure 2 and Figure 3C). The metabolic activity of *S. oralis* increased moderately at 1.95 mg/L *R. palmatum* extract compared to control biofilms, but decreased with increasing concentrations, to approximately the level of the control biofilms (Figure 3E).

Significantly decreased biofilm volumes were observed for *P. gingivalis* under the influence of 1.95 and 500 mg/L *R. palmatum* root extract (Figure 3B). As detectable with the Bacterial Viability Kit, the number of viable bacteria in the biofilm decreased significantly down to 0% at a concentration of 500 mg/L *R. palmatum* (Figure 2 and Figure 3D). Metabolic activity of *P. gingivalis* biofilms increased consistently with increasing concentrations of *R. palmatum* extract; at 31.25 and 500 mg/L *R. palmatum*, metabolic activity increased significantly compared to control (Figure 3F).

## 4. Discussion

Treatment of periodontitis and peri-implantitis is not straightforward: Bacteria embedded in biofilms are less susceptible for host defense mechanisms, or to treatment with antiseptics or antibiotics [39,40]. Multiple properties specific to the biofilm lead to specific resistance mechanisms, one of which is the failure of molecules to penetrate deeply into biofilm due to diffusion problems [41]. Furthermore, bacteria embedded in biofilms may develop antimicrobial drug resistance due to repeated antibiotic therapy. Substances such as CHX, which unselectively target bacteria, keep the total bacterial count low and may therefore lead to easier reinfection by pathogenic bacteria [21]. 

As antiseptics or antibiotics may have negative side effects, and bearing in mind the spread of antibiotic resistance, there is great interest in developing novel therapeutics against bacterial infections. Plants have shown antimicrobial activities and therefore may be a source for the development of novel antimicrobial therapeutics or may be used as co-therapeutics in combination with antibiotics in order to overcome antibiotic resistance [42]. Plant extracts may possess different biologically active compounds, which act in different target sites of microorganisms [42,43]. This may contribute to the overall activity of the plant extract and thus may slow down the development of resistance, since not only one single mechanism is involved. In order to investigate natural alternatives, our study aimed to test *R. palmatum* root extract against one commensal and one oral pathogenic species-*S. oralis* and *P. gingivalis*-initially planktonic and subsequently in biofilms. 

*R. palmatum* root extract significantly inhibited planktonic growth of *S. oralis* starting at a concentration of 62.5 mg/L, whereas planktonic growth of *P. gingivalis* was significantly inhibited at a concentration of 15.625 mg/L. Intense suppression of *P. gingivalis* planktonic growth even started at a concentration of 3.9 mg/L *R. palmatum* root extract, which is consistent with the results of the previous study [29]. These results were surprising, since plant secondary metabolites often show a higher level of activity against Gram-positive than Gram-negative bacteria [44,45]. Trans-envelope multidrug efflux pumps (EP) span the two membranes of Gram-negative bacteria [46] and are capable of extruding toxins across the membrane. *P. gingivalis* ATCC 33277 has gene clusters encoding for such efflux pumps [47]. It is known that a deactivation of efflux pumps increases antimicrobial activity in Gram-negative bacteria. Depending on the species, a 100–2000-fold antimicrobial potentiation was found for the anthraquinone rhein, a major component of *R. palmatum*, if efflux pump inhibitors were used at the same time [45]. In a previous study by the authors, HPLC analysis was performed and a comparison of the UV signals of *R. palmatum* root extract with the UV signals of the anthraquinone rhein revealed a concentration of 5% rhein in the extract [29]. Besides rhein, *R. palmatum* root extract contains phenolic components such as resveratrol [48] which is known to block efflux pumps in Gram-negative bacteria [49]. This additional antimicrobial effect may enhance antimicrobial activity against the Gram-negative *P. gingivalis*.

The metabolic activity of *S. oralis* increased significantly from a *R. palmatum* extract concentration of 31.25 mg/L, whereas the metabolic activity of *P. gingivalis* was significantly reduced at concentration at or above of 31.25 mg/L. In general, bacterial growth and metabolism are mutually related. In growing cells, the metabolic activity is high whereas it is low in non-growing or growth-inhibited cells [50]. The results for *P. gingivalis* reflect these facts. However, under the influence of higher concentrations of *R. palmatum* root extract, *S. oralis* shows increasing metabolic activity, with at the same time decreasing optical density. Antibiotics or antiseptics are known to cause bacterial stress responses and metabolic adaptions [51,52,53]. The increase in metabolic activity as a stress response is a well-known finding in previous studies [54,55]. Bacteria can stop their cell division under stress. This phenomenon, in which bacteria are metabolically active, but do not multiply, is called “quiescence” [56]. Thus, *S. oralis* seems to be slightly stressed but not completely inhibited in growth by the *R. palmatum* root extract.

*R. palmatum* root extract concentrations of 1.95, 31.25 and 500 mg/L were chosen for the subsequent biofilm experiments, in which 24-h-old biofilms were exposed to the substances for two hours. The concentration of 1.95 mg/L was selected because there were no significant effects on optical density or metabolic activity for any of the species in the planktonic experiments. A concentration of 31.25 mg/L was chosen because this gave significant differences in the bacterial growth and metabolic activity in planktonic *P. gingivalis* but not in *S. oralis*, and 500 mg/L was picked as this was the highest tested concentration in the planktonic experiments that showed significant efficacy in both species.

Evaluation of the CLMS images of *S. oralis* and *P. gingivalis* biofilms demonstrated that with an increase in *R. palmatum* root extract concentration, the biofilms of both bacterial species became more porous and holey, and the biofilm thickness increased considerably. This indicates that the *R. palmatum* root extract seems to disrupting the biofilms. A disrupting effect on biofilm structures was already detected for anthraquinone-2-carboxlic acid and rhein [57], both components of *R. palmatum*.

Biofilm volumes were analysed to determine the potential of *R. palmatum* root extract for biofilm eradication. The biofilm volumes of *S. oralis* increased slightly and the *P. gingivalis* biofilm volumes decreased slightly under the influence of *R. palmatum* root extract. There was no clear dependence on the applied concentrations of *R. palmatum* root extract. Based on these results, 1.95, 31.25 and 500 mg/L *R. palmatum* root extract appear to only slightly influence the biofilm volumes of *S. oralis* and *P. gingivalis*. Since the calculation of the biofilm volume is based on the volume of the fluorescence stained bacteria within the biofilms, the fluctuations in the biofilm volumes are not correlated to the increase in biofilm thickness. These effects could be associated with stress effects and the volumes of single cells [58], but this would have to be clarified by further experiments with single bacterial cells.

Live/dead staining was performed to evaluate the effect of *R. palmatum* root extract on bacterial viability within the biofilms in relation to membrane integrity. A concentration of 1.95 mg/L inhibited the number of viable cells in the 24-h old biofilms of *S. oralis* and *P. gingivalis* significantly after two hours of incubation. A concentration of 1.95 mg/L significantly reduced viable *P. gingivalis* from 50.38% (±10.32) to 18.59% (±14.92), whereas viable *S. oralis* was reduced from 56.82% (±9.75), to 31.72% (±7.38). At a concentration of 500 mg/L *R. palmatum* root extract, the number of viable *P. gingivalis* decreased significantly down to 0%. Similar results were found for CHX, where the MIC and minimum bactericidal concentration in *P. gingivalis* biofilms were both 4 μg/mL [59]. The results demonstrate that *R. palmatum* root extract has a strong effect on cell membrane integrity, probably due to cell membrane destruction, and thus on the vitality of bacteria in the biofilm. *P. gingivalis* was much more strongly affected than *S. oralis*.

In a study with methicillin-resistant *Staphylococcus aureus*, antibacterial and antibiofilm activity of rhein has been investigated. Rhein showed bactericidal activity (MIC < 200 µg/mL) and antibiofilm activity after 24 h of exposure, a slight biomass reduction has been observed and viable cells within the biofilm have been significantly dispersed [57]. These results are in line with our findings.

Aqueous rhubarb extract with a rhein content of 2.6% among other anthraquinones, has been reported to inhibit the biofilm formation of *Streptococcus suis* not by bacterial cell death but by downregulating transductions systems and affecting levels of transcriptional regulating factors and DNA binding proteins [60]. This may represent an additional antibiofilm mechanism of action.

The metabolic activity in *S. oralis* biofilms increased moderately at 1.95 mg/L *R. palmatum* root extract compared to control biofilms but decreased with increasing concentration, approximately to the level of the control biofilms. In contrast, metabolic activity of *P. gingivalis* biofilms increased consistently with increasing *R. palmatum* root extract concentration. Therefore, *R. palmatum* root extract had an ambiguous effect on the metabolic activity of *S. oralis*, whereas *P. gingivalis* responded with severe stress.

In summary, *R. palmatum* root extract strongly inhibited growth of planktonic *P. gingivalis*, while not inhibiting growth of the commensal *S. oralis* at identical concentrations. Furthermore, it exhibited stronger antimicrobial activity on biofilms of *P. gingivalis* than on *S. oralis*. These results indicate that *R. palmatum* root extract has the potential to selectively inhibit the oral pathogen *P. gingivalis*.

*R. palmatum* root extract slightly broke up the biofilms and made them porous. This porosity effect could potentially be useful for better penetration of active substances into biofilms. *S. oralis* showed a slight increase in metabolic activity, both in planktonic samples and in biofilms. This may be beneficial, since *S. oralis* is not known to cause cellular damage to host tissue, but might instead positively influence soft tissue cells [37]. Besides analysing the underlying mechanism of the different antibacterial efficacy of *R. palmatum*, future studies could also address the question of whether the slightly increased metabolic activity of *S. oralis* is also beneficial for host tissue homeostasis.

We are aware that clinical isolates may exhibit increased virulence and resistance mechanisms [61,62] and thus, antimicrobial results may vary among different strains. Furthermore, the effect of *R. palmatum* root extract on a wide variety of commensal and pathogen oral bacteria and polymicrobial biofilms remains to be elucidated. This specific therapeutic strategy follows the idea of the novel antibiotic amixicile, which only targets anaerobic bacteria, including *P. gingivalis, T. forsythia, F. nucleatum* and *Prevotella intermedia* [21] and thus promotes restoration of microbial homeostasis.

*R. palmatum* was not cytotoxic to oral epithelial cells in concentrations up to 500 mg/mL [30]. Thus, *R. palmatum* root extract could be used as mouthwash or toothpaste ingredient to prevent overgrowth of pathogenic bacteria, in turn, to prevent onset or progression of periodontitis. The specific and selective antibacterial mode of action of *R. palmatum* root extract would be advantageous to restore microbial homeostasis.

## 5. Conclusions

Our studies showed that *R. palmatum* root extract specifically inhibits *P. gingivalis* growth of planktonic bacteria and in biofilms in low concentrations, while growth of the oral commensal *S. oralis* was not inhibited at the same low concentrations. *R. palmatum* root extract could therefore be used topically to prevent microbiome alterations due to overgrowth of pathogenic bacteria and thus, to prevent the onset or progression of periodontitis.

## Figures and Tables

**Figure 1 microorganisms-10-00965-f001:**
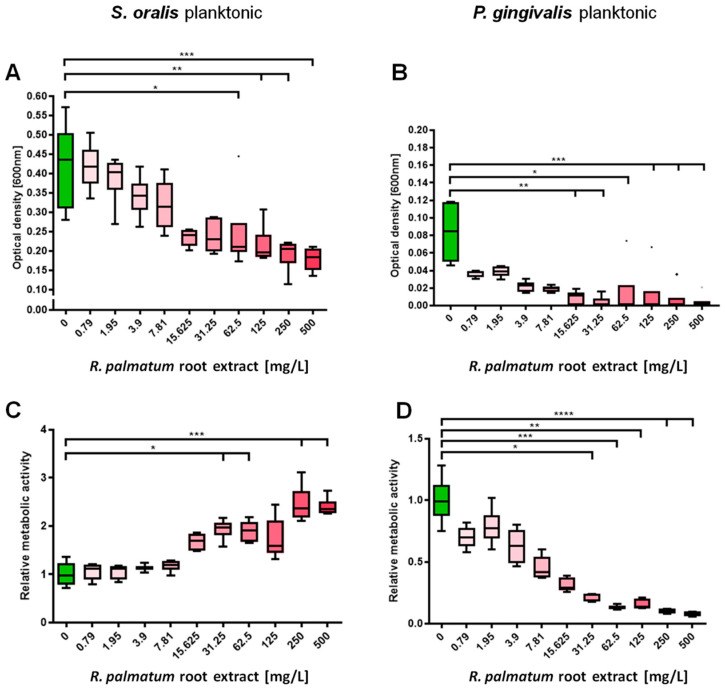
Effects of *R. palmatum* root extract on planktonic *S. oralis* and *P. gingivalis*. After 24 h of incubation. planktonic growth of *P. gingivalis* was significantly inhibited by lower concentrations of *R. palmatum* root extract than the growth of *S. oralis* was. (**A**) Treatment with 62.5 mg/L *R. palmatum* root extract or more resulted in significant reductions in the optical density of *S. oralis*. (**B**) Treatment with 15.625 mg/L *R. palmatum* root extract or more resulted in a significant reduction in the optical density of *P. gingivalis*. (**C**) Treatment with 31.25 mg/L *R. palmatum* root extract or more resulted in a significant increase in the relative metabolic activity of *S. oralis*. (**D**) Treatment with 31.25 mg/L *R. palmatum* root extract or more resulted in a significant reduction in the relative metabolic activity of *P. gingivalis*. Kruskal–Wallis test with Dunn’s correction for multiple comparisons, N = 6 with *S. oralis* strain ATCC 9811 and *P. gingivalis* strain ATCC 33277/DSM 20709. * *p* < 0.05, ** *p* < 0.01, *** *p* < 0.001, **** *p* < 0.0001.

**Figure 2 microorganisms-10-00965-f002:**
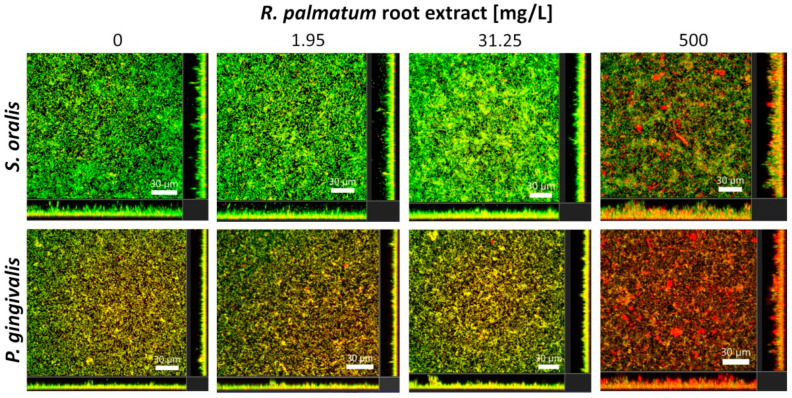
Representative CLSM images of biofilms formed by *S. oralis* and *P. gingivalis* after 24 h of cultivation followed by two hours treatment with different concentrations of *R. palmatum* root extract. The large panels show the horizontal sections (x-y planes) of the biofilms and the bottom/side panels display the vertical sections (x-z and y-z planes) through the biofilms. Biofilms were stained with SYTO 9 and propidium iodide (viable bacteria = green; dead bacteria = red or orange). The scale bars indicate 30 µm.

**Figure 3 microorganisms-10-00965-f003:**
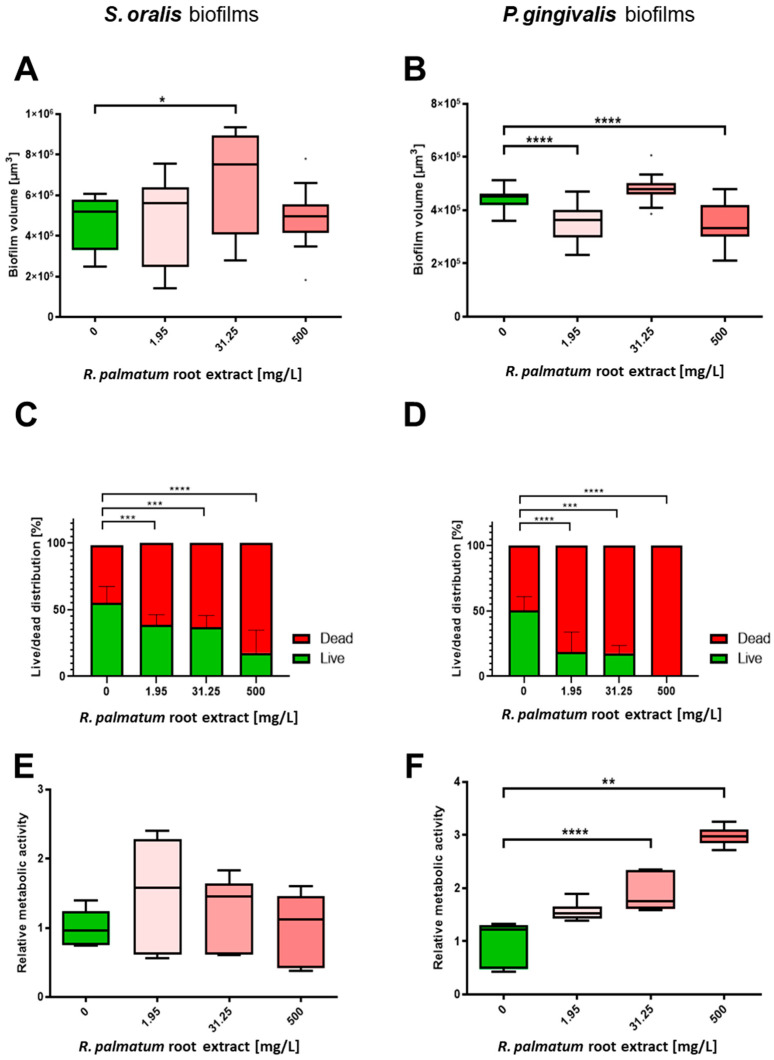
Effects of *R. palmatum* root extract on *S. oralis* and *P. gingivalis* biofilms. After two hours of incubation, viability of *S. oralis* and *P. gingivalis* in 24-h biofilms was significantly inhibited by low concentrations of *R. palmatum* root extract of at least 1.95 mg/L-with a greater decrease in the viability of *P. gingivalis*. (**A**) Treatment with 1.95 mg/L and 31.25 mg/L *R. palmatum* root extract increased biofilm volume of *S. oralis*, whereas biofilm volume was decreased by 500 mg/L *R. palmatum* root extract. (**B**) Treatment with 1.95 mg/L and 500 mg/L *R. palmatum* root extract significantly decreased biofilm volume of *P. gingivalis*. (**C**) Treatment with 1.95 mg/L or more of *R. palmatum* root extract significantly decreased viability of *S. oralis* in biofilms. (**D**) Treatment with 1.95 mg/L or more of *R. palmatum* root extract significantly decreased viability of *P. gingivalis* in biofilms. (**E**) Treatment with 1.95 mg/L and above of *R. palmatum* root extract increased metabolic activity of *S. oralis* in biofilms. (**F**) Treatment with 1.95 mg/L or more of *R. palmatum* root extract increased metabolic activity of *P. gingivalis* in biofilms. N = 6 with *S. oralis* strain ATCC 9811 and *P. gingivalis* strain ATCC 33277/DSM 20709. * *p* < 0.05, ** *p* < 0.01, *** *p* < 0.001, **** *p* < 0.0001.

**Table 1 microorganisms-10-00965-t001:** Final concentrations of *R. palmatum* root extract.

*R. palmatum* Root Extract										
Concentration (mg/L)	0.79	1.95	3.90	7.8125	15.625	31.25	62.5	125	250	500

## Data Availability

Data are contained in this manuscript or Appendix A.

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
