# Peer review of "Antiplanktonic and Antibiofilm Activity of Rheum palmatum Against Streptococcus oralis and Porphyromonas gingivalis"

_microorganisms, 2022, doi:10.3390/microorganisms10050965_

Round 1

Reviewer 1 Report

1.what is the major compound in the plant extract 

2.author should be add the HPLC peaks and list out in the table  compounds and properties 

3.author should be add the live and dead baclight  assay clsm images of main text 

4.author should be add the biofilm images 

5.what is the biofilm thickness of bacteria  control and after treatments?? author should be add the biofilm thickness CLSM ortho or 3D images in biofilm thickness to calculate to comstat  and add the results in main text

6.what is the antibiofilm mechanism of plant extract 

7.author should be add the  images of plant ,plant extracts, and add the mathodology of plant collections,plant distribution,how to confirm the plant,any botanical vocher specimen???

8.what is the positive control of antibiofilm activity??

9.why author not tested the CHX or Povidine or any antibiotic for positive control test  -author clearly justify 

10.author how to this experiments in to commericialise in to mouth wash ??

Author Response

Dear Reviewer, 

Reviewer 1: what is the major compound in the plant extract 

Response: The method applied here is not sufficient to fully answer this question, as it was mainly intended to provide the possibility of comparison between different R. palmatum samples. As anthraquinones have been described to be the main bioactive compound in R. palmatum, we analyzed the sample qualitatively by HPLC-MS (figure S1 was added to the SI) for their presence as free anthraquinones and anthraquinone glycosides, but other compounds of this extract were not further separated and purified. As rhein has been described as the most potent member, we compared our sample with a rhein standard as described in the previous study (1).

Reviewer 1: author should be add the HPLC peaks and list out in the table compounds and properties 

Response: In section 2.3, the sentence “The results of the HPLC analysis have been published in a previous study [29].” has been altered to “The results of the HPLC analysis containing the UV chromatogram at 254 nm and the corresponding peak list have been published in a previous study [29].” for the readers’ convenience. 

The HPLC study was mainly performed to be able to compare the extract used in this study with other extracts at a later stage (HPLC fingerprinting). For this reason, and because we already knew about the outstanding biological activity of rhein on P. gingivalis based on our previous study, other compounds of this extract were not further separated and purified, but the anthraquinone derivates of R. palmatum root extract were quantified via HPLC analysis and figure S1 was added to the SI. 

The following sentence has been added to the Materials and Methods section: “The UV chromatogram at 254 nm and the extracted ion current chromatograms (EIC) of the five most abundant anthraquinones and their glycosides in negative ionization mode (ESI – ) is depicted in figure S1.” (page 03, line 123-125).

Reviewer 1: author should be add the live and dead baclight  assay clsm images of main text 

Response: Representative CLSM images of live dead stained biofilms were added to the results part. Please see figure 02 page 8.  

Reviewer 1: author should be add the biofilm images 

Response:  Representative CLSM images of live dead stained biofilms were added to the results part. Please see figure 02 page 8.  

Reviewer 1: what is the biofilm thickness of bacteria control and after treatments?? author should be add the biofilm thickness CLSM ortho or 3D images in biofilm thickness to calculate to comstat and add the results in main text

Response: We thank you for your suggestion to observe the biofilm thickness. Unfortunately, we are not able to calculate the exact biofilm thickness/height with the Imaris software, as it does not offer this tool. However, we examined each biofilm regarding its biofilm height and added orthogonal views of the representative biofilms with XY, YZ, and XZ planes to the results part (Figure 02 page 8). In fact, there were exciting differences between the untreated and treated samples. We saw a clear increase in biofilm thickness/height and the biofilm structures became more porous and holey under the treatment of R. palmatum for both species. The following sentences has been added to the Results section:

“One representative CLSM image for each experiment of S. oralis and P. gingivalis biofilms after 24 hours of cultivation followed by 2 hours treatment with R. palmatum root extract is shown in figure 2. With increasing concentration of R. palmatum root extract, the proportion of dead bacteria within the biofilms increased visually for both species; however, the antibacterial effect seemed to be much stronger for P. gingivalis than for S. oralis. For both bacterial species, the vertical sections showed a considerable increase in biofilm thickness and the biofilm structures became more porous and holey, as R. palmatum concentration increased – in particular at 500 mg/L.“ (p 06, line 225-232).

Reviewer 1. :what is the antibiofilm mechanism of plant extract 

Response: Primarily, our study showed that the application of R. palmatum root resulted in bacterial cell death.  Furthermore, biofilm structures became more porous and holey, suggesting that the biofilms became disrupted on the surface. This is another very important finding in addition to the antimicrobial effect. This biofilm disruption may result in a better penetration rate of antimicrobial substances. Nevertheless, the underlying mechanism was not analysed by the authors. The following sentence was added to the discussion section: “Evaluation of the CLMS images of S. oralis and P. gingivalis biofilms demonstrated that with an increase in R. palmatum root extract concentration, the biofilms of both bacterial species became more porous and holey, and the biofilm thickness increased considerably. This indicates that the R. palmatum root extract seems to disrupting the biofilms. A disrupting effect on biofilm structures was already detected for anthraquinone-2-carboxlic acid and rhein [57], both components of R. palmatum.”(page 09 line 335-340).

Reviewer 1: author should be add the  images of plant ,plant extracts, and add the mathodology of plant collections, plant distribution, how to confirm the plant,any botanical vocher specimen???

Response: The identity of the R. palmatum root extract used in this study was identified according to the European Pharmacopeia and the European Union herbal monograph on Rheum palmatum L. and Rheum officinale Baillon, radix (EMEA/HMPC/189624/2007). In order to clarify the identity, the following sentence was added to the material and methods section “The identity of the R. palmatum root extract used in this study was identified according to the European Pharmacopeia and the European Union herbal monograph on Rheum palmatum L. and Rheum officinale Baillon, radix (EMEA/HMPC/189624/2007)”. (page 03 line 116-119). The authors will grant access to specification documents for the editor.

Reviewer 1: what is the positive control of antibiofilm activity??

Response: Our test system worked with two types of controls: negative/sterility controls (growth media only and the plant extract in the growth media) and positive growth controls (bacteria without any treatment) and a validation is therefore performed by positive and negative controls within the system. No reference substance was included to the tests.

Reviewer 1: why author not tested the CHX or Povidine or any antibiotic for positive control test  - author clearly justify 

Response: Our test system itself worked with two types of controls: negative/sterility controls (growth media only and the plant extract in the growth media) and positive growth controls (bacteria without any treatment) and a validation is therefore performed by positive and negative controls within the system. In a previous study, we proceeded similarly (https://pubmed.ncbi.nlm.nih.gov/33794846/). Ethanol was only included as a control, because it is an ingredient of REPHA-OS and it should be avoided that the antimicrobial effect was caused by the alcohol. In this study, the aim was not to compare the antimicrobial effect with a known reference such as PVP-Iod or CHX. Furthermore, the antimicrobial effects of R. palmatum root extract were verified with commercial kits, whose mechanism basically does not need to be validated. Currently no standard guidelines exist regarding reference substances in antibiofilm testing, as it is for example the case in antibiotic susceptibility tests. We thank the reviewer for the suggestion and agree that a comparison of the new specific antimicrobial approach with PVP-Iod, CHX and possibly oral antibiotics would make sense. The authors suggest comparative tests with clinical multispecies isolates and isolate strains for further studies. 

Reviewer 1: author how to this experiments in to commericialise in to mouth wash ?

Response: The following subclause was added to the conclusion section: “Our studies showed that R. palmatum root extract specifically inhibited P. gingivalis growth of planktonic bacteria and in biofilms in low concentrations, while growth of the oral commensal S. oralis was not inhibited at the same low concentrations. R. palmatum root extract could therefore be used topically to prevent microbiome alterations due to overgrowth of pathogenic bacteria and thus, to prevent the onset or progression of periodontitis (page 10 line 409 - 410).

Regarding the commercial use, the authors declared one conflict of interest.

  • Müller-Heupt LK, Vierengel N, Groß J, Opatz T, Deschner J, von Loewenich FD. Antimicrobial Activity of Eucalyptus globulus, Azadirachta indica, Glycyrrhiza glabra, Rheum palmatumExtracts and Rhein against Porphyromonas gingivalis. Antibiotics (Basel). 2022 Jan 31;11(2):186. doi: 10.3390/antibiotics11020186. PMID: 35203789; PMCID: PMC8868162.

Best regards,

Reviewer 2 Report

My comments are attached in the revision file.

Author Response

Dear Reviewer, 

Reviewer 2: The authors compared the effect of Rheum palmatum root extract against the pathogenic strain of P. gingivalis and the commensal species S. oralis. They also studied anti-planktonic, metabolic and antibiofilm activity.

Why do the authors perform an HPLC on the root extract and then do not test the compounds corresponding to individual peaks?

Response: The HPLC study was mainly performed to be able to compare the extract used in this study with other extracts at a later stage (HPLC fingerprinting). For this reason, and because we already knew that the outstanding biological activity of the R. palmatum extract on P. gingivalis could be attributed largely to rhein based on our previous study, other compounds of this extract were not further separated and purified, but the anthraquinone derivates of R. palmatum root extract were analysed by HPLC and HPLC-MS and figure S1 was added to the SI. 

We were able to demonstrate earlier that the activity of the extract mainly corresponds to its rhein content (which is also the lead component, the equivalent content of which is specified in the commercial extract as 4%) so that other components, although being present in much larger amounts, do not seem to contribute to this particular type of biological activity. 

Reviewer 2: In order to complete the study, the authors should test the HPLC separated compounds for their antiplanktonic ability, their metabolism and anti-biofilm on both bacterial strains.

Response: We thank the reviewer for this comment. However, in the previously reported study, the R. palmatum root extract was analysed on an analytical HPLC system. Therefore, it was not possible to collect the individual components in quantities with which the corresponding tests can be carried out. 

Also, as mentioned in “section 4. Discussion”, the aim of this study was to investigate the effect of the extract due to several benefits of extracts with multiple compounds compared to a single compound. Clinical application of only a single compound against P. gingivalis and S. oralis may lead to a faster formation of antimicrobial resistances. Furthermore, pure anthraquinones are only fully soluble in alkaline media and therefore considered as cytotoxic and not usable as an ingredient to oral hygiene products. Therefore, the main objective was to analyse the full spectrum extract of R. palmatum. A complete compound analysis could be the subject of further studies.

Reviewer 2: Did the authors try to test whether R. palmatum root extract prevents biofilm formation?

Response: We thank the reviewer for this comment. The authors did not test whether R. palmatum root extract prevents biofilm formation since substances for oral hygiene are usually rinsed off very quickly and thus, do not have a longer lasting effect which may prevent novel biofilm formation.

Best regards,

Round 2

Reviewer 1 Report

Accept

Reviewer 2 Report

The answers provided by the authors are satisfactory.